

Regulating effects of mixed cultivated grasslands in surface water conservation and soil erosion
reduction along with restoration of alpine degraded hillsides
Yulei Ma[1], Yu Liu[1,2], Jesús Rodrigo-Comino[3], Manuel López-Vicente[4], Gao-Lin Wu[1,2,5]
*[1] State Key Laboratory of Soil Erosion and Dryland Farming on the Loess Plateau, Institute of Soil*
*and Water Conservation, Northwest A & F University, Yangling, Shaanxi 712100, China*
*[2] Institute of Soil and Water Conservation, Chinese Academy of Sciences and Ministry of Water*
*Resource, Yangling, Shaanxi 712100, China*
*[3] Departamento de Análisis Geográfico Regional y Geografía Física, Facultad de Filosofía y Letras,*
*Campus Universitario de Cartuja, University of Granada, Granada, Spain*
*[d] AQUATERRA Research Group, CICA-UDC, Universidade da Coruña. As Carballeiras s/n, Campus*
*de Elviña, A Coruña 15071, Spain*
*[5] CAS Center for Excellence in Quaternary Science and Global Change, Xi'an, 710061, China*
**Correspondence**: wugaolin@nwsuaf.edu.cn (G.L. Wu).
**Correspondence address:** State Key Laboratory of Soil Erosion and Dryland Farming on the Loess
Plateau, Northwest A & F University, No 26, Xinong Road, Yangling, Shaanxi 712100, P.R. China
Phone: +86- (29) 87012884 Fax: +86- (29) 87016082



**ABSTRACT**
Vegetation restoration is one of the most effective measures to control runoff and sediment by human
management. Nevertheless, few studies have been undertaken to objectively analyze the effectiveness
of the effects of plant restoration on regional water availability, especially, in mixed-cultivated
grasslands in alpine degraded hillsides. In this research, we carried out *in situ* monitoring using micro-
plots to investigate the impact of three strategies, combining two grass species per plot (three species
in total), in a 20-degree slope on the activation and volume of surface runoff and soil loss in alpine
degraded hillsides for three years (2019, 2020 and 2022). A bare-soil plot was used as control. The
findings indicated that mixed-cultivated grasslands can effectively conserve water and decrease soil
loss along the increasing planting ages. Grass community of *Deschampsia cespitosa* and *Poa*
*pratensis L.cv.* Qinghai was the most effective in reducing soil erosion. From 2019 to 2022, the values
of the runoff reduction ratio decreased for *Deschampsia cespitosa* and *Elymus nutans* (*DE*), *Poa*
*pratensis L.cv.* Qinghai and *Elymus nutans* (*PE*), and *Poa pratensis L.cv.* Qinghai and *Deschampsia*
*cespitosa* and (*PD*) from -79.3% to -115.4%, from -130.4% to -156.1%, and from -48.5% to -87.6%,
respectively. On the contrary, the mean soil erosion reduction ratio of the cultivated grass
communities increased from -184.8% to 18.0% (in *DE*), from -231.5% to 24.3% (in *PE*), and from -
139.3% to 31.9% (in *PD*), respectively, from 2019 to 2022; and the corresponding mean values of
sediment concentration reduction ratio also increased from -120.9% to 55.8% (in *DE*), -from 112.4%
to 59.7% (in *PE*), and from -94.3% to 62.1% (in *PD*). This implied that protection measures should
be considered a priority during the initial planting stage of cultivated grassland in alpine degraded
hillsides. The key factors affecting soil loss and runoff were rainfall amount, duration and intensity
(60-min intensity). We conclude that the results of this study can serve as scientific guides to design
efficient policy decisions for planning the most effective vegetation restoration in the severely





degraded hillside alpine grasslands.

***Keywords:*** Alpine grassland; Degraded hillside; Mixed-cultivated grassland; land management; runoff; soil erosion.

## 1    Introduction

Grasslands are an essential component of terrestrial ecosystems and one of the regions with the highest concentration of human activity (O'Mara, 2012). Grasslands contribute significantly to biodiversity maintenance, climate mitigation, carbon sequestration, and water supply and regulation (Bardgett et al., 2021). Despite the importance of grasslands, about half of them are degraded globally, with 5% undergoing severe degradation, which has become a major issue for humanity to overcome (Gang et al., 2014; Török et al., 2021). To date, considerable studies have been conducted to analyze the root causes, negative impacts and restoration measures of grassland degradation (Gang et al., 2014; Grman et al., 2021; Han et al., 2020). Water and soil are critical for human survival and development, as well as irreplaceable basic natural resources that maintain the function of natural ecosystems and the development of socioeconomic systems. Few studies, however, have particularly examined how well-restored grasslands can regulate water supply and prevent soil erosion (Minea et al., 2022). This is particularly important for alpine grasslands, which play a vital role in the supply of fresh water and the development of livestock husbandry (Cui et al., 2022), and thus, it is necessary to assess the efficiency of grassland restoration in maintaining runoff and protecting soil.

Vegetation restoration is universally viewed as one of the most effective ways to control runoff and sediment around the world (Anache et al., 2018). The effects of vegetation cover properties on runoff and soil loss reduction are strongly connected to plant types, leaf and branch coverage, above-



ground biomass, litter biomass, and root systems (Liu et al., 2022; Freschet and Roumet, 2017;
Gyssels et al., 2005; Zhu et al., 2021). Furthermore, the processes of runoff and soil loss are
significantly influenced by the enhancement of soil characteristics with the growth of vegetation
(Schwarz et al., 2015; Gyssels et al., 2005). Although vegetation restoration has the potential to be a
key method of environmental restoration under human management, the sustainability of local
economic and environmental development is negatively impacted by the inappropriate selection of
species (Hoek Van Dijke et al., 2022). For example, cultivated grasslands were already advocated as
a sensible solution for the conservation of soil and water, as well as the regrowth of vegetation in
semi-arid mountain areas (Liu et al., 2022; Wu et al., 2010). Grasses community with multiple
stratified structures is better at conserving water and decreasing soil loss than that with a single
composition and structure (Mohammad and Adam, 2010).
Soil erosion can decrease with grasses above- and below-ground, biomass grasses plant and litter
cover, as well as root systems (De Baets et al., 2007). Grasslands can control water erosion relying
on the role of aboveground biomass in dissipating flow energy (Bochet and García-Fayos, 2004),
living roots in topsoil resistance against concentrated runoff flow that activates soil loss (Zhang et al.,
2013), grass plant cover in intercepting rainfall (Liu et al., 2019), and litter cover in enhancing
rainwater infiltration (Liu et al., 2022). Moreover, the interaction of soil and rich grassroots can
remarkably alter the physical properties of topsoil, thereby enhancing its resistance to erosion
(Schwarz et al., 2015; Wang et al., 2018). The impact of grassroots on the characteristics of soil could
be summed up as follows: i) increasing the stability of soil aggregates through aggregating fine soil
particles into solid macroaggregates; ii) enhancing soil cohesiveness through interweaving with the
soil; and iii) changing soil bulk density through reinforcing soil mass (Wang et al., 2018; Gyssels et
al., 2005). For example, numerous recent studies have confirmed that a shallow yet dense fibrous root



system appears to be more effective at controlling water erosion (Liu et al., 2022; De Baets et al.,

86    2007).

Especially, alpine grasslands are the predominant plant type in the Qinghai-Tibetan Plateau,
accounting for 44% and 6% of total grassland areas in China and the world, respectively (Wang et
al., 2016). Alpine grasslands are fragile ecosystems when rapid changes are involved and due to
climate change and non-planned human activities have suffered substantial degradation in recent
decades. This situation is leading to a drop in vegetation cover and an increase in bare surfaces,
especially for hillsides grassland, ultimately posing a great hazard to the plateau from water and soil
loss (Fig.1) (Liu et al., 2022). The Qinghai-Tibetan Plateau is the headwaters for many of Asia's major
rivers (Xu, 2018). Long-term and widespread degradation of hillside alpine grassland has changed
the soil water balance, reducing runoff, which in turn lower river streamflow and ultimately limits the
sustainable development of local and downstream regions. The establishment of artificial grassland
on severely degraded hillsides offered the dual benefits of boosting productivity and improving the
ecological environment of alpine grasslands (Shang et al., 2008; Liu et al., 2022).
Despite previous reports have been focused on carbon sequestration capacity, vegetation
characteristics, soil quality and productivity of cultivated grassland (Wang et al., 2013; Wen et al.,
2018), few studies have focused on the impacts of artificial grassland on the provision of runoff and
prevention of soil erosion on the alpine hillsides. Here, we present novel research to examine the
ability of alpine hillsides cultivated grasslands to regulate runoff and soil loss through three different
mixed artificial grasslands compared to degraded bare land in alpine degraded hillsides by a three-
year field experiment. In this vein, this study has realistic implications for understanding the
contribution of artificial grasslands restoration on soil erosion control in the degraded alpine hillside.





## 2   Materials and methods

### 2.1   Study area

This study was carried out in the representative area of Zhique Village (33°40′01″ N and 99°43′06″ E, elevation over 4200 m a.s.l), Dari County, Qinghai province, which served as a field experimental site for the restoration of degraded alpine grassland in the Three Rivers on the Qinghai-Tibetan Plateau (Fig. 1). The climate conditions correspond to a typical highland one with low temperatures throughout the year, i.e., not showing distinct seasons, just cold and warm ones. The study region has an average annual temperature of -0.6 °C and an average annual precipitation of 513 mm (Li et al., 2012). Nevertheless, the majority of the precipitation and the warm season falls during the vegetation growth period (from May to September), favoring optimal conditions for the development of vegetation. The soil type in the study area is classified as alpine meadow soil (IUSS-WRB, 2015) (Liu et al., 2022). Currently, the remnant vegetation in this site is composed of an alpine shrub (*Salix cupularis* and *Potentilla fruticose*), alpine meadow (*Kobresia pygmaea, Kobresia humilis* and *Kobresia capillifoli*) and swamp meadow (*Carex atrofusca, Poa annua* and *Carex parva*). The degraded alpine grassland was restored through man-made planting or natural succession.

### 2.2   Experimental design and measurement

The importance of artificial grassland in restoring alpine degraded grassland is widely accepted (Wen et al., 2018; Wu et al., 2010). The degraded hillslopes are the main component of runoff generation and concentration areas on the Qinghai-Tibetan Plateau. Hence, the grass species chosen for artificial grasslands should not only be grazing-tolerant and good forage but also prevent soil and water loss. Potential grass species should also be fully acclimated to harsh alpine climatic and have the complementarity of morphological characteristics and living habits (Liu et al., 2022). The community



established by blending complementary grass species has a hierarchical vertical cover structure and
little inter- or intraspecific competition. Following the above-mentioned guidelines for choosing grass
species, we ultimately decided on three species (*Deschampsia cespitosa, Poa pratensis L. cv.* Qinghai
and *Elymus nutans*) from the most widely utilized grass species. *Deschampsia cespitosa* is a cool-
season bunching grass native to alpine environments. It typically forms a low, dense tussock (to 30–
50 cm tall) of very thin (0.5 cm wide), arching, flat to inrolled, dark green grass blades (to 5 cm long).
*Deschampsia cespitosa*, a common bottom grass, has 70% of its above-ground plants growing
between 0 and 30 cm tall. *Elymus nutans* is a common and important plant species in the alpine
meadows of the Qinghai-Tibetan plateau (Chen et al., 2009). It is a valuable fodder grass in alpine
locations that has been extensively employed for animal production, disturbed grassland restoration,
and artificial grassland construction due to its resilience to cold, drought and pests (Ren et al., 2010).
*Elymus nutans* is a herbaceous perennial species with sparsely tufted culms that can grow to heights
of 70 to 100 cm (Liu et al., 2022). *Poa pratensis L. cv. Qinghai* is the common and dominant species
native to the Qinghai-Tibetan Plateau. It is an excellent species that have been selected and cultivated
to restore degraded alpine grasslands. Also, *Poa pratensis L. cv.* Qinghai is a herbaceous perennial
species with erect or geniculate base culms that grow 20–60 cm tall.
To reveal the effectiveness of mixed artificial grassland in controlling runoff and soil loss on
hillsides, field observation of mixed grass plots designed by us was conducted from the 2019 to 2022
growing seasons. Therefore, one plot with bare land (as control) and three plots with two mixed
artificial grasslands per plot of *Deschampsia cespitosa* and *Elymus nutans* (*DE*), *Poa pratensis L.cv.*
Qinghai and *Elymus nutans* (*PE*), and *Poa pratensis L.cv.* Qinghai and *Deschampsia cespitosa* (*PD*)
were selected as the testing site (Fig. 1). All four plots were bounded by steel plates (30 cm high and
2 mm thick sheet) and built during May 2019, with an area of 10 m$^2$ (2 m wide and 5 m long parallel



to the maximum slope gradient). To collect solely runoff and sediment from the runoff areas, the steel
plate was put vertically into the soil to a depth of about 10 cm, with the remainder sticking out from
the soil surface. At the outlet of each plot, a steel runoff collection and calibrated tank (75 L) were set
up to gather sediment and runoff (Fig. 1). To prevent the collected runoff from being lost to
evaporation, the calibrated tank was set inside a sealed vat.

In addition, each runoff plot grass seeding was finished in May 2019. On the runoff plots, grass

seed was made to a depth of less than 1 cm in strips at 20 cm intervals following the plowing. The
seeding rate was set at 6.0 g m$^{-2}$ for *Poa pratensis L.cv.* Qinghai and *Deschampsia cespitosa* and 4.5
g m$^{-2}$ for *Elymus nutans* to ensure a constant number of plants based on germination and seedling
emergence rates. None of the runoff plots was disturbed by human activity during the observation
period (2019–2022), including grazing, harvesting, and excavation.

## 2.3    Rainfall, runoff and sediment measurement

A Vantage pro 2$^{TM}$ weather station (Davis Instruments Corp., USA) with a measurement accuracy of
4% is positioned next to the experimental plots to monitor precipitation intensity and duration (Fig.
1). A total of 42 precipitation events were recorded from 2019 to 2022 throughout the growing season.
Snow was not collected, and only rainfall was recorded. Precipitation characteristics of each event,
including amount ($P$), duration ($RD$), maximum intensities of 60 minutes ($RI_{60}$), and average intensity
($ARI$) were recorded. After each rainfall-runoff event, both runoff and sediment were collected right
away. The water level in the calibrated tank was first measured to calculate the runoff volume. Then,
runoff was fully mixed inside the calibrated tank, and two 500 ml bottles were used to obtain mixture
samples of sediment and runoff. When the calibrated tank had less than 1000 ml of runoff sample, all
runoff was collected. Lastly, the calibrated tank was cleaned in order to collect sediment and runoff





for the subsequent rainfall-runoff event. The mixture samples in the bottle were transported back to
the lab to be filtered on qualitative filter paper. The filter paper with sediment was air-dried to a
consistent weight at 105 °C. The ratio of soil loss amount to runoff volume in the mixed samples was
applied to calculate the sediment concentration. Finally, runoff volume and sediment concentration
were multiplied to calculate soil loss in each plot.
We collected runoff and sediment data during the growing season for the years 2019 to 2022. Data
for 2021 could not be collected due to the prevention and control strategies for coronavirus (COVID-
19). Soil erosion and runoff were portrayed in this work by soil erosion modulus (t km$^{-2}$) and runoff
depth (mm). The runoff depth ($R$) and soil erosion modulus ($S$) could be calculated using the following
formulas:

$$R = \frac{V_R}{A} \times 10^3 \tag{1}$$

$$S = \frac{SE}{A} \tag{2}$$

where $V_R$ is the volume of runoff (m$^3$), $SE$ is the amount of soil erosion (g), and $A$ is the area of
runoff plot (m$^2$).

**2.4   Vegetation and soil properties measurement**
Vegetation cover ($VC$) was measured from 2019 to 2022 growing seasons. After collecting runoff
samples in late August 2022, the quadrats (50 × 50 cm) were positioned in the up., mid-, and
downslope areas of each runoff event. Plant litter biomass ($LB$) was measured using oven drying and
collection techniques. Undisturbed soil samples were taken in the 0–10 cm soil layers. Soil bulk
density ($BD$) was determined using undisturbed soil samples collected by steel rings. Root mass
density ($RMD$) was obtained using a root drill, followed by washing with water and drying in oven.
The cohesiveness was calculated using soil direct shear and the Coulomb.
**2.5 Calculating the reduction effect of runoff and soil loss**
Four metrics were employed to assess the efficiencies of the mixed cultivated grasslands in regulating
runoff and soil loss, which were: The runoff reduction ratio (*RRE*, %), sediment concentration
reduction ratio (*CRE*, %), soil erosion reduction ratio (*SRE*, %), and the percentage of runoff reduction
ratio to soil loss reduction ratio (*RRSR*) (Zhu et al., 2021). High values of *RRE*, *SRE* or *CRE* indicated
that vegetation was able to reduce runoff, soil erosion or sediment concentration compared to the
rates observed in the control plot (bare land). In addition, a low *RRSR* implied that vegetation was
more beneficial in minimizing soil erosion than in minimizing runoff (Liu et al., 2020). These indices
were calculated as follows:

$$RRE = \frac{R_c - R_v}{R_c} \tag{3}$$

$$SRE = \frac{S_c - S_v}{S_c} \tag{4}$$

$$CRE = \frac{C_c - C_v}{C_c} \tag{5}$$

$$RRSR = \frac{RRE}{SRE} \tag{6}$$

where $R_c$ and $R_v$ are the runoff depths of the bare plot and plots covered by mixed-cultivated
grasslands; $S_c$ and $S_v$ are the soil erosion modulus of the bare plot and plots covered by mixed-
cultivated grasslands; $C_c$ and $C_v$ are the sediment concentrations of the bare plot and plots covered
by mixed cultivated grasslands, respectively.

**2.6 Statistical analyses**
Using SPSS statistics software (IBM, USA, version 26.0), all data were analyzed. The Kolmogorov–
Smirnov test was used to test the normality of data. Duncan's multiple range tests of one-way analysis
of variance (ANOVA) were applied to compare the significant differences in soil and vegetation
characteristics, runoff depth, soil erosion modulus, and runoff and soil loss reduction ratio under



various mixed-cultivated grasslands at 0.05 significance levels. Also, the method of path analysis was
used to identify the major factors influencing runoff and soil loss.

**3   Results**
**3.1   Runoff and soil loss under various mixed-cultivated grasslands**
Grass communities dramatically influenced runoff and soil erosion. One-way analysis of variance
(ANOVA) revealed that runoff significantly ($P < 0.05$) increased after the severely alpine degraded
hillside was restored by the mixed-cultivated grassland (Fig. 2). During the growing seasons of 2019,
2020, and 2022, the average runoff depths of bare ground were 0.23, 0.34 and 0.25 mm, respectively,
all less than the average runoff of mixed-cultivated grassland *DE* (0.44, 0.55 and 0.43 mm), *PE* (0.59,
0.51 and 0.54 mm), and *PD* (0.50 mm, 0.38 mm and 0.40 mm). However, the amount of soil loss in
grasslands was significantly influenced by the age of the planting age. As depicted in Fig. 2b, in both
2019 and 2020 (the first and second years of planting) mixed artificial grasses produced higher soil
loss than bare land, whereas mixed artificial grasses lost less soil in the fourth year of planting (2022)
than bare ground. The soil erosion of bare land (0.23 t km$^{-2}$) was 1.4, 1.3 and 1.9 times that of *DE*,
*PE* and *PD* (0.16, 0.18 and 0.12 t km$^{-2}$, respectively). The results showed that the community of *Poa*
*pratensis L. cv.* Qinghai and *Deschampsia cespitosa* seemed to be more successful at controlling soil
loss and runoff.

**3.2   Runoff and soil loss reduction under various mixed-cultivated grasslands**
Fig. 3 illustrates the runoff, soil loss and sediment concentration reduction ratio after planting various
mixed-cultivated grasslands. Lower *RRE* values indicated a better ability of surface water



conservation for grasslands, while higher *SRE* and *CRE* values indicated better effectiveness of
grasslands in soil loss reduction. The mean *RRE* values of the grass community *DE*, *PE*, and *PD* were
-79.3%, -130.4% and -48.5% in 2019, -36.9%, -53.5% and -21.5% in 2020, and -115.4%, -156.1%
and -87.6% in 2022 (Fig. 3a). Regardless of the combination of the above-mentioned grass species,
the increase ratio of runoff in 2022 was significantly higher than that in 2019 and 2020 (the first and
second years of planting). In the grass communities, the root structure had a significant influence on
the *SRE*. The *SRE* of the three mixed-cultivated grasslands (*DE*, *PE*, and *PD*) increased with
increasing planting age. It is worth noting that the average *SRE* values in the grassland communities
of *DE*, *PE* and *PD*were 18.0%, 24.3%, and 31.9% in 2022, respectively (Fig. 3b). Additionally, all
mixed-cultivated grasslands displayed a significant rise in *CRE* from the first to the fourth year. The
mean *CRE* values of the cultivated-grassland communities *DE*, *PE*, and *PD* increased from -120.9%
to 55.8%, from -112.4% to 59.7%, and from -94.3% to 62.1% from 2019 to 2022, respectively (Fig.
3c). Regardless of the age of the grasslands, the value of *RRSR* was less than 1, suggesting that the
soil erosion reduction effect of the grasslands was higher than its runoff reduction effect (Fig. 3d).

**3.3   Key factors affecting runoff and soil loss**
Precipitation characteristics and vegetation features significantly influenced the hydrological
response of the soil. Here, the path analysis was applied to identify the key elements affecting soil
loss. The results of this analysis indicated that the sum of path coefficients of $RI_{60}$, *ARI*, *RD*, *P*, *VC*
and LB were 0.31, 0.18, 0.36, 0.40, 0.32 and 0.13, respectively (Table 1). This implies that *P*, *RD*,
*VC* and $RI_{60}$ had positive effects on runoff yield, with *P* being the most crucial. The direct and indirect
path coefficients of $RI_{60}$, *ARI*, *RD*, *P*, *VC* and LB were 0.24, 0.37, 0.67, -0.18, 0.29, -0.12 and 0.07, -
0.19, -0.31, 0.57, 0.03, 0.25, respectively (Table 1). These findings revealed that the impact factors



of *ARI* and *RD* were primarily responsible for their direct influences, whereas the impact factors of *P*
and *LB* were mainly attributed to their indirect influences. For instance, *P*, in combination with other
factors, particularly $RI_{60}$ and *RD*, contributed significantly to runoff.
Soil loss was significantly influenced by *R*, $RI_{60}$, *RD* and *P*, with being *R* the most relevant. The
sum of path coefficients of *R*, $RI_{60}$, *RD* and *P* were 0.51, -0.14, -0.16 and 0.12, respectively (Table 2).
These results show that *R* and *P* had a promotional effect, whereas $RI_{60}$ and *RD* had an inhibitory
effect on soil loss. Meanwhile, *R* and *P* had a direct positive influence on soil erosion, with direct
path coefficients of 0.60 and 0.28, whereas $RI_{60}$ and *RD* had a direct negative influence on soil erosion,
with direct path coefficients of -0.29 and -0.41 (Table 2). In addition, the direct and indirect path
coefficients both indicated that *LB* had an inhibitory influence on the soil erosion modulus, with
values of -0.10 and -0.03, respectively.
**4 Discussion**
**4.1 Contribution of cultivated grasslands on soil conservation and runoff maintenance**
The mixed-cultivated grasslands (*DE*, *PE*, and *PD*) were able to effectively conserve water,
improving soil water retention and infiltration indirectly, and minimize soil loss (Fig. 3). This finding
is similar to those of studies conducted checking different grassland communities (Liu et al., 2019;
Liu et al., 2022). The soil erosion modulus of all three mixed-cultivated grasslands (*DE*, *PE* and *PD*)
was higher than that of the bare ground in the first and second years following planting, but in the
fourth year, the bare ground had a higher soil erosion modulus than the three mixed-cultivated
grasslands (Fig. 2). The changes in soil erosion were dominantly attributed to the developing of the
root system and improvement of soil structure (Zhu et al., 2021). The loosening of the soil structure
caused by the seeding method of plowing resulted in a greater soil erosion modulus of the three





mixed-cultivated grasslands than the bare ground at the beginning of the planting. We confirmed that
the age of plantation was a key factor to understand the inter-annual changes of soil erosion. This idea
was also demonstrated in other types of primary land uses such as woody crops or young forests
(Rodrigo-Comino, et al., 2018). Nevertheless, we hypothesize that grassland topsoil demonstrated a
stronger resilience to erosion as its root system grew, which had a reinforcement impact on the soil
and led to lower soil loss in the fourth year of planting than that of the bare land. The topsoil (0-10
cm) of the grasslands had significantly different soil properties from the bare land in the fourth year
after planting, as detailed in Table 3. In comparison to $BL$, the root mass density and soil cohesion of
grasslands $DE$, $PE$ and $PD$ increased by 400.0%, 428.4% and 459.8%, and by 67.0%, 53.8% and
92.7%, respectively.
The mixed-cultivated grasslands significantly increased surface runoff when compared with bare
land in this study. The grasslands in this study had more abundant fibrous roots in the surface soil
compared with bare land (Table 3). The dense and compact sward formed by surface soil interwoven
with fibrous roots and soil particles cemented by root secretions limited the timely infiltration of
rainfall, ultimately resulting in increased runoff (Niu et al., 2021; Gyssels et al., 2005).

**4.2    Effect of rainfall and grassland community characteristics on runoff and soil loss**
Changes in precipitation regime and vegetation cover significantly influence the process of surface
runoff and soil erosion, such as dynamic changes in runoff depth and soil erosion rate (Mohamadi
and Kavian, 2015b; Bochet et al., 2006). In this study, the $VC$ had a directly promoted effect on
surface runoff. Moreover, this result was in line with the finding of Niu et al. (2021), who reported
that the surface runoff increased with the grassland coverage. Our results also indicate that $P$ could
have an indirect effect on surface runoff via $RD$ and $RI_{60}$. This suggests that heavier and longer-lasting



rainfall events were more conducive to the generation of surface runoff (Dos Santos et al., 2017). The
findings demonstrated that $R$ and $ARI$ were the most and second most influential factors for promoting
the occurrence of soil erosion (Table 2). The primary cause for this is that runoff velocity increases
with rising precipitation intensity (Wang et al., 2013), which probably further enhanced the capacity
of soil detachment and transport by surface runoff (Zhu et al., 2021). Furthermore, $LB$ influenced soil
loss directly and negatively (Table 2), indicating that the effectiveness of grasslands in reducing soil
loss increased as litter biomass increased. Liu et al. (2022) found that the soil loss rate decreased with
the increase of litter biomass in the grassland. The plant litter could intercept precipitation, reducing
rainfall kinetic energy and splash erosion, and increasing surface roughness (Liu et al., 2017; Xia et
al., 2019); all these processes favored a reduction in the rates of runoff yield and soil loss.

**4.3    Implications for artificial grasslands restoration on the degraded alpine hillside**
Our findings demonstrated that mixed-cultivated grasslands with complementing morphological
features and habits can be more effective at conserving water and reducing soil erosion. The
combination community of *Poapratensis L.cv.* Qinghai and *Deschampsia cespitosa* (*PD*) exhibited
an effective role in controlling soil loss on the degraded alpine hillside. The community of *PD* was
much better than the communities of *PE* and *DE* in reducing soil loss (Fig. 3), which could likely be
due to two reasons. First, the morphological characteristics of *Deschampsia cespitosa*, *Poapratensis*
*L.cv.* Qinghai and *Elymus nutans* were dense clump type, rhizomatic-sparse clump type, and sparse
clump perennial grasses, respectively. The mix of dense (*Deschampsia cespitosa*) and sparse
(*Poapratensis L.cv.* Qinghai) grasses can complement each other morphologically and structurally,
thereby more effectively reducing the kinetic energy of raindrops (Liu et al., 2022). *Poapratensis L.cv.*
Qinghai, a rhizomatic grass, also has abundant root systems intertwined with the soil, increasing soil



cohesion and consequently reducing soil detachment capacity (Wang et al., 2018). A comparison of
the three mixed artificial types of grass in Table 3 revealed that grassland *PD* had the highest soil
cohesion (8.92 kPa). However, at the start of planting, the mixed planted grassland had a greater soil
erosion modulus than bare ground, whereas the function of reducing soil loss was reached in the 4th
year of planting (Figs. 2 and 3). This suggested that protection measures, such as mesh covering and
anti-trampling, may be taken into account to reduce soil loss in the initial planting stage of cultivated
grassland in alpine hillsides (Liu et al., 2022). Moreover, grass may also be planted with a no-till
system to avoid the initial increase of soil erosion at the initial phases of cultivated grassland by
destroying soil structure (Karayel and Sarauskis, 2019).

Cultivated grasslands, as a crucial component of vegetation restoration, have been widely

employed to rehabilitate degraded alpine hillsides (Shang et al., 2008). Nevertheless, plant restoration
is not necessarily beneficial to the long-term viability of on- and off-site ecosystems' functions,
including natural succession and river ecosystems. The selected vegetation types ought to be
advantageous for the ecosystem's sustainability, both on- and off-site, such as maintaining river
streamflow and unrestricted natural succession. The seed prices of cultivated grass communities of
*Deschampsia cespitosa* and *Elymus nutans*, *Poa pratensis L.cv.* Qinghai and *Elymus nutans*, and *Poa*
*pratensis L.cv.* Qinghai and *Deschampsia cespitosa* were about \$690, \$750 and \$480 per ha. Planting
properly cultivated grassland on the alpine degraded hillsides can achieve both environmental and
economic benefits. The Qinghai-Tibetan Plateau has traditionally been referred the third pole and the
"world's water tower", playing a significant and unique role in the global climate and energy-water
cycle (Xu, 2018). Many of Asia's major rivers, including the Yellow, Changjiang, Mekong, Ganges
and Indus Rivers, originate from the Qinghai-Tibetan Plateau. Hence, restored vegetation ecosystems
should contribute a large quantity of clear and high-quality water resources. This study proved that





mixed-cultivated grasslands could conserve water and decrease soil loss, and thus, reduce overland
flow turbidity.
**5  Conclusions**
Based on the measured data during the 2019, 2020 and 2022 growing seasons, our findings showed
that mixed-cultivated grassland could effectively conserve surface water and decrease soil loss, which
could better contribute to the functions of maintaining better surface water resources and reducing
sediment yield on severely degraded hillside alpine grasslands. To guarantee that they can perform
the aforementioned functions, mixed-cultivated grasslands need protection measures in the initial
planting stage. Our results also suggested that mixed-cultivated grasslands with complementary
morphology and structure, such as the mixture of the dense clump (*Deschampsia cespitosa*) and
rhizomatic-sparse clump (*Poa pratensis L.cv.* Qinghai), could be more effective in maintaining
surface runoff and reducing sediment. Precipitation amount, duration, vegetation coverage and
maximum 60-minute intensity were the predominant factors affecting surface runoff and soil loss.
The erosion resistance contribution of the above-ground community characteristics and below-ground
roots along the cultivated time could maintain a relatively high surface runoff and decrease sediment
production. These findings have potential implications for understanding the contribution of artificial
grasslands restoration on soil erosion control in the degraded hillsides of alpine areas.

*Data availability.* All data needed to evaluate the conclusions in the paper are present in the paper.

*Author contributions.* Yulei Ma: Investigation, Formal analysis, Methodology, Software, Writing -
original draft. Yu Liu: Investigation, Methodology, Project administration. Jesús Rodrigo-Comino:



Interpretation of data, Writing - review & editing. Manuel López-Vicente: Interpretation of data,
Writing - review & editing. Gao-Lin Wu: Conceptualization, Funding acquisition, Supervision,
Writing - original draft, review & editing.

*Competing interests.* The authors declare that they have no known competing financial interests or
personal relationships that could have appeared to influence the work reported in this paper.

*Disclaimer.* Publisher's note: Copernicus Publications remains neutral with regard to jurisdictional
claims in published maps and institutional affiliations
*Acknowledgments.* We thank Yi-Fan Liu for help of the data analysis, and thank Li-Rong Zhao and
Jia-Xin Qian for their help in the field investigation.
*Financial support.* This research was funded by the National Natural Science Foundation of China
(NSFC41930755, NSFC41907058, NSFC32230068), the Strategic Priority Research Program of the
Chinese Academy of Sciences (XDB40000000), and the Second Stage's Research and Technique
Extending Project of Sanjiangyuan Ecological Protection and Building in Qinghai (2019-S-1).

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





**Table 1**. Results of path analysis of the factors affecting runoff depth.

| Influence factor | Direct path coefficient | Indirect path coefficient | | | | | | | Sum of path coefficient |
|---|---|---|---|---|---|---|---|---|---|
| | | $RI_{60}$ | $ARI$ | $RD$ | $P$ | $VC$ | $LB$ | Total | |
| $RI_{60}$ | 0.24* | | 0.25 | -0.09 | -0.11 | 0.02 | 0.00 | 0.07 | 0.31 |
| $ARI$ | 0.37** | 0.16 | | -0.34 | -0.05 | 0.02 | 0.02 | -0.19 | 0.18 |
| $RD$ | 0.67** | -0.03 | -0.18 | | -0.08 | 0.03 | -0.03 | -0.31 | 0.36 |
| $P$ | -0.18** | 0.14 | 0.10 | 0.31 | | 0.02 | 0.00 | 0.57 | 0.40 |
| $VC$ | 0.29** | 0.01 | 0.03 | 0.06 | -0.01 | | -0.06 | 0.03 | 0.32 |
| $LB$ | -0.12 | 0.01 | -0.09 | 0.18 | 0.00 | 0.15 | | 0.25 | 0.13 |

Note: $RI_{60}$ is maximum 60-minute intensity (mm h$^{-1}$), $ARI$ is average intensity (mm h$^{-1}$), $RD$ is rainfall
duration (h), $P$ is rainfall amount (mm), $VC$ is vegetation coverage (%), $LB$ is litter biomass (g m$^{-2}$).
** means the correlation is significant at 0.01 significance level.



**Table 2.** Results of path analysis of the factors affecting soil erosion modulus.

| Influence factor | Direct path coefficient | Indirect path coefficient | | | | | | | | Sum of path coefficient |
|---|---|---|---|---|---|---|---|---|---|---|
| | | $R$ | $RI_{60}$ | $ARI$ | $RD$ | $P$ | $VC$ | $LB$ | Total | |
| $R$ | 0.60** | | -0.10 | 0.01 | -0.08 | 0.08 | 0.01 | -0.01 | -0.09 | 0.51 |
| $RI_{60}$ | -0.29** | 0.20 | | 0.02 | -0.22 | 0.15 | 0.00 | 0.00 | 0.16 | -0.13 |
| $ARI$ | 0.04 | 0.14 | -0.19 | | 0.22 | 0.06 | 0.00 | 0.01 | 0.25 | 0.28 |
| $RD$ | -0.41** | 0.12 | 0.05 | -0.02 | | 0.13 | 0.00 | -0.03 | 0.26 | -0.15 |
| $P$ | 0.28** | 0.18 | -0.16 | 0.01 | -0.19 | | 0.00 | 0.00 | -0.17 | 0.11 |
| $VC$ | 0.03 | 0.16 | -0.02 | 0.00 | -0.02 | 0.01 | | -0.05 | 0.07 | 0.10 |
| $LB$ | -0.10 | 0.07 | -0.01 | -0.01 | -0.10 | 0.01 | 0.01 | | -0.02 | -0.12 |

Note: $R$ is surface runoff (mm), $RI_{60}$ is maximum 60-minute intensity (mm h$^{-1}$), $ARI$ is average intensity (mm h$^{-1}$), $RD$ is rainfall duration (h), $P$ is rainfall amount (mm), $VC$ is vegetation coverage (%), LB is litter biomass (g m$^{-2}$). * means the correlation is significant at 0.05 significance level, and ** means the correlation is significant at 0.01 significance level.





**Table 3.** Topsoil characteristics of four-years-old mixed-cultivated grasslands.

| Mixed cultivated grasslands | Soil depth (cm) | Bulk density (g cm⁻³) | Soil saturated water content (%) | Field capacity (%) | Total porosity (%) | Root mass density (kg m⁻³) | soil cohesion (kPa) |
|---|---|---|---|---|---|---|---|
| *DE* | | 1.19±0.07[a] | 44.93 0.04[a] | 33.72±0.01[a] | 55.23±0.03[a] | 10.20±2.55[a] | 7.73±3.85[a] |
| *PE* | | 1.15±0.01[a] | 50.16±0.05[a] | 35.19±0.02[a] | 58.79±0.03[a] | 10.78±3.54[a] | 7.12±1.98[a] |
| *PD* | 0–10 | 1.19±0.04[a] | 46.85±0.06[a] | 34.81±0.04[a] | 56.4±0.03[a] | 11.42±4.92[a] | 8.92±0.86[a] |
| *BL* | | 1.26±0.07[a] | 40.09±0.04[a] | 31.51±0.01[b] | 57.39±0.04[a] | 2.04±1.51[b] | 4.63±3.55[a] |

Note: Different lowercase letters indicate soil characteristics differed significant between different
grasslands ($p < 0.05$). Values given represent mean values ± standard deviation. The same letter in
the same column means that differences are not significant at $p = 0.05$.

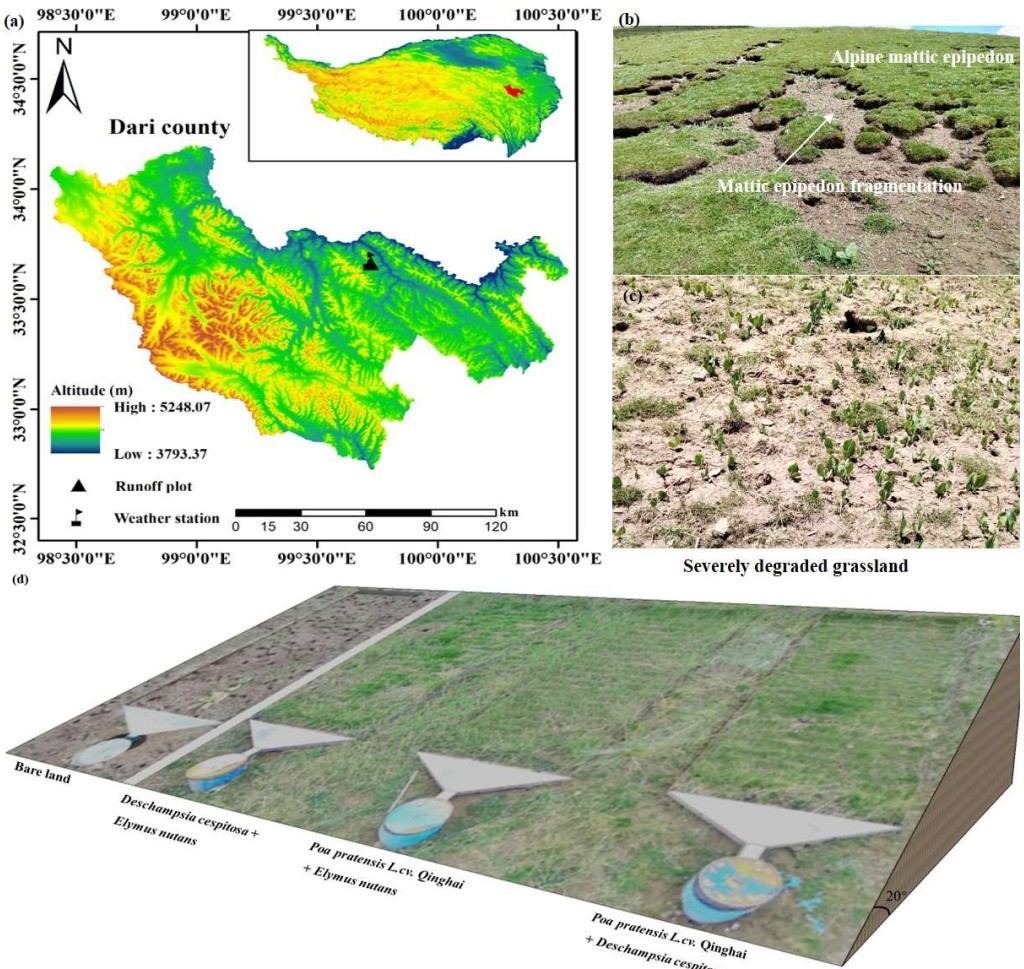

**Figure 1.** The location of the study area on the Qinghai-Tibetan Plateau, and the location of runoff

plots in the study area. (a) The location of the study area, (b) the fragmenting mattic epipedom on the

alpine hillslope and (c) severely degraded grassland (bare land) formed by the disappearance of mattic

eppipedom and (d) four runoff plots on bare ground and mixed-cultivated grasslands. A typical badly

deteriorated grassland with a slope of 20° was selected to plant mixed grasses. Runoff plots were

photographed with a drone in the early stages of the 2022 growing season.



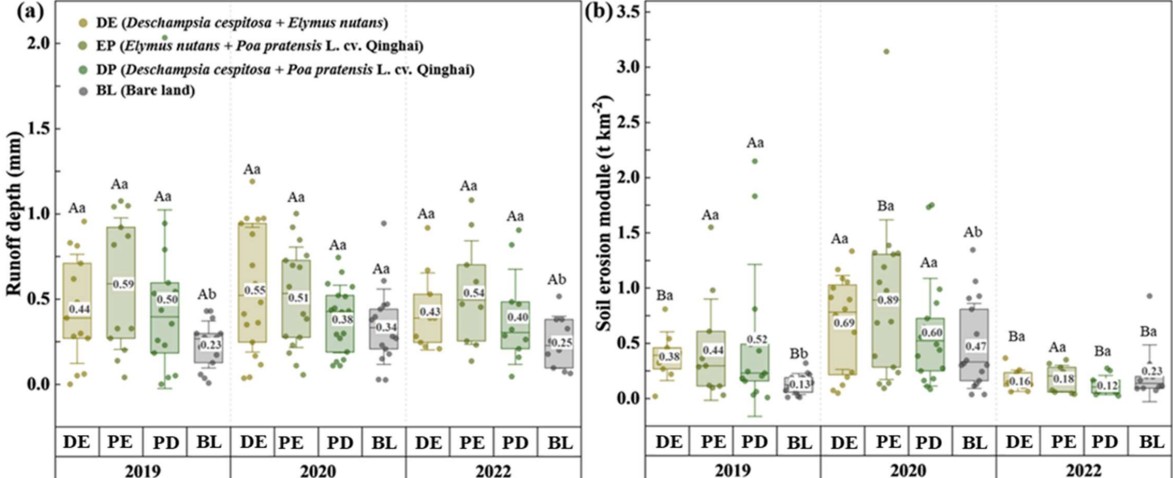

**Figure 2.** Changes in soil erosion and runoff under various mixed-cultivated grasslands from 2019 to

2022. (a) Runoff depth and (b) soil erosion module. Note: Different capital letters mean that

differences were significant in different years for the same grassland community, and different

lowercase letters mean that differences were significant between different communities in the same

year.

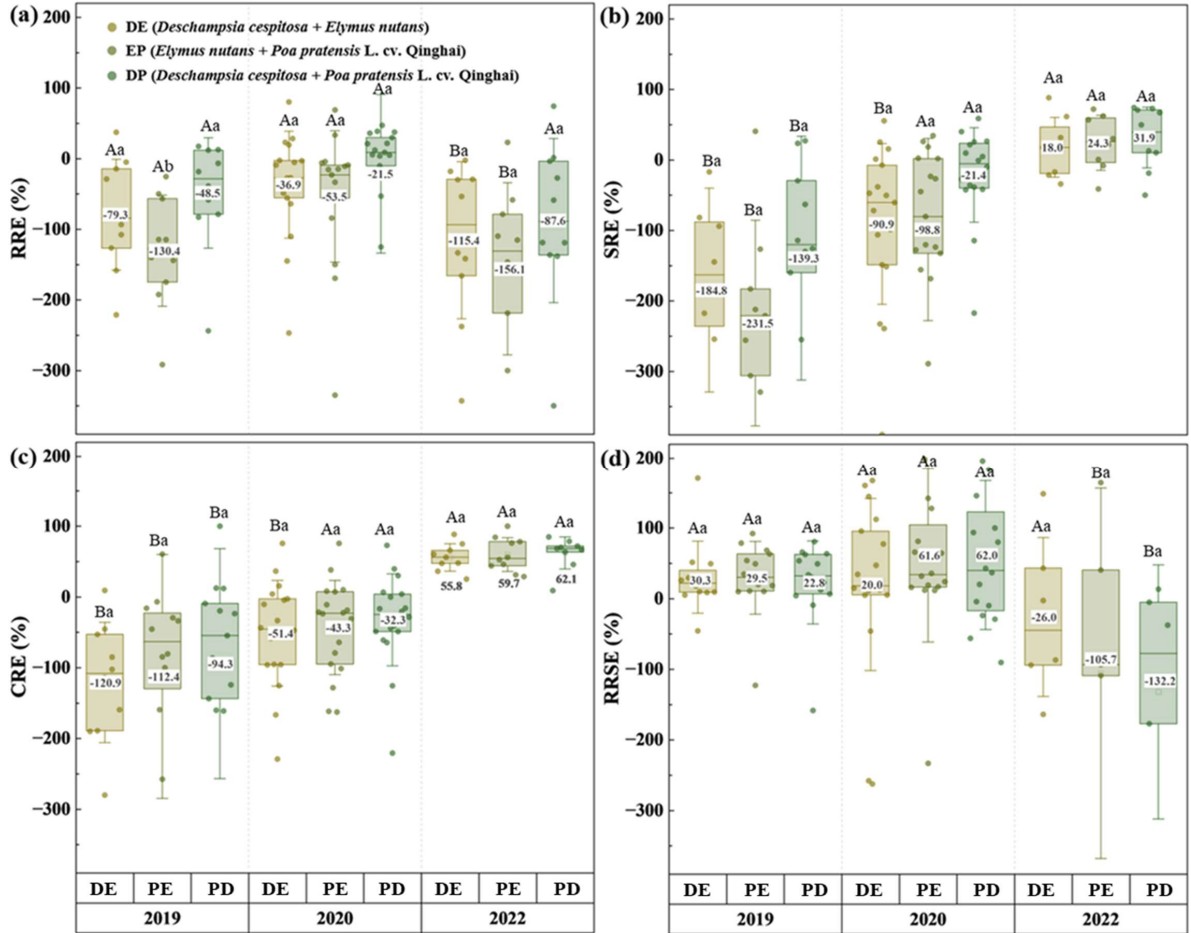

**Figure 3.** Runoff, soil loss and sediment concentration reduction ratio under different mixed-cultivated grasslands from 2019 to 2022. (a) Runoff reduction ratio (*RRE*), (b) soil loss reduction ratio (*SRE*), (c) sediment concentration reduction ratio (*CRE*) and (d) the percent of runoff reduction ratio to soil loss reduction ratio (*RRSR*). Note: Different capital letters mean that differences were significant in different years for the same grassland community, and different lowercase letters mean that differences were significant between different communities in the same year.