# Peer review of "Regulating effects of mixed cultivated grasslands in surface water conservation and soil erosion"

_Hydrology and Earth System Sciences, 2023_

## Author Comment (AC1)

**Initial response to CC1**

*Qianjin Liu (Community 1)'s original text in black with our initial response in* blue.

This manuscript presented an important information on the change of soil and water loss in the mixed-cultivated grass land in alpine degraded hillsides, and the influencing factors were identified and interpreted in a detail. A minor revision needs to be performed before acceptance.

The detailed comments and suggestions are listed as follows:

We thank Qianjin Liu for his constructive comments on our manuscript.

Line 45: Please add the main human activities.

The main human activities are grazing, and this information will be added in the resubmitted manuscript.

Line 59: considered may be better than viewed.

The word "viewed" will been replaced with "considered" in the resubmitted manuscript.

Line 76: Please use promote instead of active.

The wording "living roots in topsoil resistance against concentrated runoff flow that activates soil loss (Zhang et al., 2013)" will be replaced by "living roots in decreasing soil detachment capacity by runoff (Zhang et al., 2013)" in the resubmitted manuscript.

Line 82: Please delete solid.

The word "solid" will be deleted in the resubmitted manuscript.

Line 104: Please provide a statement on mixed artificial grasslands and on temporal variations in soil and water loss after planning or seeding at other regimes.

The following information will be added to the end of the paragraph in the resubmitted manuscript: "This study aimed to (1) assess the temporal variations in soil and water loss of *DE, PE* and *PD* grasslands during the growing season and under natural rainfall; and (2) determine the key factors influencing the mixed-cultivated grasslands in controlling runoff and soil erosion."

Line 125-126: Please move these kind of contents to the instruction section.

The sentence will be moved to the Introduction section in the resubmitted manuscript.

Line 154: solely?

This should have been "only" and will be corrected in the resubmitted manuscript.

Line 355: please give the temporal change of soil and water loss and the main reason.

The following information will be added at the end of the paragraph in the resubmitted manuscript: "Planting the mixed-cultivated grasslands after ploughing loosened the soil structure and thus increased sediment concentration in runoff during the first stage after planting. Subsequently, sediment concentration decreased with the growth of the root system of the mixed-cultivated grasslands, strengthening the sloping soils due to the root architecture."

Fig. 2b: Please have a check the capital letter for PE.

We have carefully checked and will revise the capital letter in **Figure 2** in the resubmitted manuscript.

---

## Author Comment (AC2)

**Initial response to RC1**

*Corinna Gall (Reviewer 1)'s original text in black with our initial response in* blue.

In this study, surface runoff and soil erosion are measured for three different combinations of grass species and a bare soil control sample in degraded alpine hillsides on the Qinghai-Tibetan Plateau, using runoff plots under natural rainfall. The goal is to find a grass mixture for grassland restoration that is best adapted to the needs of the ecosystem. This is a very important approach to grassland restoration that has not been sufficiently studied. Therefore, I consider this study worth publishing within the scope of HESS, however, it needs to be considerably revised in terms of comprehensibility.

Unfortunately, it is difficult to follow the structure in the introduction and in some parts to understand the reasoning in the discussion. In the introduction, it is challenging to find a common thread. In general, the introduction needs to be modified in order to lead the reader specifically to the goal or hypotheses of the study. I propose to explain the problems regarding surface runoff and soil erosion in this alpine grassland ecosystem in more detail at the beginning before pointing out the specific research gaps. Also, it would be good to end the introduction with hypotheses that you will Response throughout the manuscript. In the discussion, it is not always clear how the conclusions are reached. Some statements are too strong on the basis of the data presented and should be considered in a more differentiated way.

Additionally, I have the following comments and questions for the authors to consider:

We thank Corinna Gall for the positive words about our manuscript and constructive feedback and suggestions that will help refine our paper.

**Abstract**

Line 19: Please rephrase "effectiveness of the effects".

This should have been "the effects" and will be updated in the resubmitted manuscript.

Lines 21-22: Are you sure the size is still micro-runoff plots?

This should have been "runoff plots" and will be updated in the resubmitted manuscript.

Lines 21-24: If you measured surface runoff and soil loss, what can you say about the regional water availability mentioned above? It seems to me here that the representation of the knowledge gap and the research objective do not fit together properly.

Vegetation restoration is probably not always positive for the sustainability of hydrological effects, such as soil water conditions, runoff and evapotranspiration, and in some cases due to the excessive water comsupmtion by plants that lead to soil drying in some soil layers. The Qinghai-Tibetan Plateau is the headwater for many of Asia's major rivers. Hence, it is necessary to evaluate whether vegetation restoration maintains or undercuts the gains of downstream water resources. To avoid confusing the reader, we have revised the sentence as follows: "Nevertheless, few studies have been undertaken to analyze the effects of plant restoration on maintaining the stability of the hydrological cycle, especially, in alpine degraded hillsides where mixed-cultivated grasslands predominate in the landscape."

Line 25: What is meant here by "conserve water"?

This should have been "maintain runoff" and will be updated in the resubmitted manuscript.

Lines 30-35: There are too much values for the abstract section. Maybe only list significant changes.

We will only list significant changes in the Abstract.

**Introduction**

Line 58: What do you mean with "maintaining runoff" here? Wouldn't it be desirable to reduce runoff and promote water retention in the soil?

Thank you for your valuable suggestion. "Maintaining runoff" in this study refers to a relatively lower runoff reduction effect. Large-scale vegetation restoration generally reduces the local runoff coefficient (ratio between the amount of runoff to the amount of rainfall depth), which is not conducive to the health of river ecosystems, especially in arid and semi-arid regions.

Line 59: I recommend "considered" instead of "viewed".

The word "viewed" will be replaced by "considered" in the resubmitted manuscript.

Line 61: I recommend "plant species" instead of "plant types".

The word "plant types" will be replaced by "plant species" in the resubmitted manuscript.

Line 68: This reference deals with tree restoration. Does this statement also apply to grasslands? Please provide a reference.

Thank you for this observation. We will erase the tree restoration reference and add more specific references for grassland restoration like Huang et al. (2017; 2019).

Line 70: Please change to "Grass communities … are …".

We will rephrase this sentence in the resubmission as follows: "Grasses communities with multiple stratified structures are better at maintaining surface water and decreasing soil loss than that with a single composition and structure (Mohammad and Adam, 2010)."

Lines 70-72: What do you mean with "conserve water" in this context? Maybe use "retain" instead?

The word "conserve water" will be replaced by "maintain surface water" in the resubmitted manuscript.

Line 73: Please rephrase "biomass grasses plant and litter cover".

This should have been "grasses above- and below-ground biomass and litter cover" and the text will be updated in the resubmitted manuscript.

Lines 74-78: This sentence is difficult to understand. Please consider rephrasing.

We will rephrase this sentence as follows: "Grasslands can control water erosion relying on the role of the aboveground biomass in dissipating flow energy (Bochet and García-Fayos, 2004), living roots in decreasing soil detachment capacity (Zhang et al., 2013), grass plant cover in intercepting rainfall (Liu et al., 2019), and litter cover in enhancing rainwater infiltration (Liu et al., 2022)."

Line 83: Please specify "changing" here.

The word "changing" will be replaced by "decreasing" in the resubmitted manuscript.

Lines 84-86: A dense root system is more effective than what? Please clarify.

The sentence will appear rewritten as follows: "For example, numerous recent studies have confirmed that a grass with shallow yet dense fibrous root system appears to be more effective at controlling

water erosion than a grass with good ground cover but low root density (Liu et al., 2022; De Baets et al., 2007; Bochet et al., 2006)."

Line 87: I would suggest referring to alpine grassland as an ecosystem or landscape unit rather than a plant type.

The words "plant type" will be replaced by "ecosystem".

Lines 94-96: This sentence explains my question in Line 58. This explanation must be given earlier.

Precipitation is the main water source in semi-arid areas and the conversion of precipitation to runoff is influenced by vegetation restoration, which in turn influences river flow recharge. To avoid misleading the reader, this information will be inserted before line 58 of the introduction section in the resubmitted manuscript.

**Methods**

Line 110: It is unclear what the term "representative area" refers to.

Zhique Village is the model area to study the artificial restoration of the severely degraded meadows (https://www.sohu.com/a/489177164_362042). The term "representative area" will be replaced by "model area" in the resubmitted manuscript.

Line 112: What means "Three Rivers" here? Is it a landscape unit or a district?

The study area is the headwater for many of Asia's major rivers, including Yangtze, Yellow and Lancang River in China. Therefore, the study area is also known as the Three Rivers Source in China. To avoid misleading the reader, we have erased the words "in the Three Rivers"in the resubmitted manuscript.

Lines 113-118: Please be more specific here. Perhaps provide additional averages for temperature and precipitation per season to illustrate the differences between warm and cold seasons.

We will add the following specific information in the resubmitted manuscripts: "In the study region, the average annual temperature is -0.1°C, with monthly variations from -18.3°C in January to 12.4°C in July (Li et al., 2018). The average annual precipitation is 416 mm, with the majority of it falling from July to September."

Line 129: Please use "climate" instead of "climatic".

The word "climatic" will be replaced by "climate" in the resubmitted manuscript.

Lines 129-130: I would suggest to change the wording to "…have complementary morphological characteristics and habits".

The wording will change to "have complementary morphological characteristics and habits" in the resubmitted manuscript.

Line 149: Figure 1 shows that the control plot is not completely bare, but is degraded grassland. Please mention this in the text as well.

Thank you for this comment. We will carefully checke the whole text and replace the expression "bare land" by "severely degraded meadows" in the resubmitted manuscript.

Line 154: Please use "runoff plot" instead of "runoff area".

This should have been "runoff plot" and will be updated in the resubmitted manuscript.

Lines 93 +113 + 152 + 157: Please make clear which part of Figure 1 you are referring to (a, b, c, or d).

Thank you for noticing this. We will indicate this in the resubmitted manuscript.

Line 169: Does this mean that all precipitation events outside the growing season were snow and therefore measurements were only taken during the growing season? How did you deal with melt water erosion in the time between the growing seasons?

There are only two distinct seasons in the study area —cold and warm— due to its high altitude (average elevation of 4000 m above sea level). Hence, solid precipitation is mainly distributed in the cold season, whereas rainfall (85.2% of the annual precipitation) is distributed in the warm season (i.e., plant-growing season, which ranges from May to September) (Shi et al., 2020; Ren et al., 2010; Zheng et al., 2022). Meltwater erosion is an important type of soil erosion in the Qinghai-Tibet region, which is very serious from March to May due to the soil freeze-thaw effect and low vegetation cover (Zheng et al., 2022; Shi et al., 2020). Our experiment were conducted during the peak growing season, and little snowmelt erosion occurred in the study area during this period. Additionally, spring

meltwater is one of the major drivers of soil erosion in alpine meadows, and further studies of meltwater erosion will be conducted in the future.

Lines 174-175: How did you ensure that the water and sediment in the tank were evenly mixed in order to collect two representative 500-ml samples?

To ensure that the water and sediment were uniformly mixed in the tank, we thoroughly swirled with a stirring bar.

Lines 178-179: I think there is an incorrect description of the drying of the sediment here. Air drying to a constant weight at 105°C does not fit together.

This is a typo, and the word "air-dried" will be replaced by "oven-dried" in the resubmitted manuscript.

Line 184-188: In my opinion, scaling to km² is not appropriate for such small runoff plots. I would recommend expressing soil erosion in kg / g m$^{-2}$. Moreover, it is confusing to extrapolate only soil erosion and not surface runoff within the same sample.

Thank you for your valuable suggestion. After each rainfall-runoff event, both runoff and sediment were collected right away. The water level in the calibrated tank was first measured to calculate the runoff volume. Then, runoff was fully mixed inside the calibrated tank using a stirring bar to thoroughly whirl, and two 500 ml bottles were used to obtain mixture samples of sediment and runoff. We will expresse soil erosion in g m$^{-2}$ in the resubmitted manuscript.

Lines 191-193: It is not clear how and when vegetation cover and plant litter biomass were obtained. Was the vegetation coverage estimated for August 2022 only or after each rainfall event? What do you mean by "collection techniques" in the context of plant litter biomass? Was the plant litter biomass determined randomly in the 50x50cm frames or for the entire plot? Please specify and provide references for your methods.

Vegetation cover (VC) was measured monthly from 2019 to 2022 growing seasons using a steel wire frame (50 cm × 50 cm) subdivided into 25 plots of 10 cm × 10 cm. After collecting runoff samples each year, the quadrats (50 cm × 50 cm) were positioned in the up., -, mid-, and down-slope areas in

late August 2022. Litter in each quadrant was collected and oven-dried to determine litter biomass (LB) (Zhu et al., 2021). This information will be added in the methods in the resubmitted manuscript.

Line 202: Please provide the primary source for these indices. They were also used in Zhao et al. 2014. The dynamic effects of pastures and crop on runoff and sediments reduction at loess slopes under simulated rainfall conditions.

The primary source for the indices (RRE, SRE, CRE and *RRSR*) will be added in the resubmitted manuscript.

Lines 207-210: Be sure to use a consistent spelling of "mixed cultivated grassland". Sometimes you use it with a dash, sometimes without.

We have carefully checked the whole paper, and will replace "mixed cultivated grassland" by "mixed-cultivated grassland" in the resubmitted manuscript.

Line 215: I suggest the expression "to test for significant differences between".

This will be updated in the resubmitted manuscript.

Figure 1: Please use your abbreviations for the treatments also in Figure 1. The runoff plots appear a bit distorted and of different sizes in the illustration, perhaps a simple top view would be more appropriate here. Please check the spelling and punctuation in the figure caption again (line 541-546).

Thank you for these recommendations. All treatment descriptions will be abbreviated and explained in the figure caption, and the illustration in Figure 1 has also been replaced with a top view. We will update Fig. 1 to reflect this change in the resubmitted manuscript.

[Figure]

Figure 1. Location of the study area on the Qinghai-Tibetan Plateau, and location of the runoff plots in the study area (a). The fragmenting mattic epipedom on the alpine hillslope (b) and the severely degraded meadows formed by the disappearance of mattic eppipedom (c). Runoff plots of severely degraded meadows (SEM) and mixed-cultivated grasslands (d). A typical severely degraded meadows with a slope of 20° was selected to plant mixed grasses. Runoff plots were photographed with a drone in the early stages of the 2022 growing season. DE, *Deschampsia cespitosa* and *Elymus nutans*; PE, *Poa pratensis L.cv.* Qinghai and *Elymus nutans*; and PD, *Poa pratensis L.cv.* Qinghai and *Deschampsia cespitosa*.

**Results**

Line 225: Please be consistent with the designation of your treatments. Usually you have used the term "bare land" or the abbreviation "BL".

As control plots were not completely bare, but rather severely degraded meadows, we will replace the expression "bare land" by "severely degraded meadow", and abbreviation will be "*SDM*" in the resubmitted manuscript.

Lines 221-234: It would be very valuable here to additionally provide a diagram with the values of vegetation cover per year and plot. Especially, it would be good to know the vegetation cover of the bare land treatment, as it was not completely bare. In addition, information is needed on how you managed the bare land treatment during the three-year experiment. If there were no human impacts as described in the methods, surely the vegetation cover on the bare land was lower in 2019 than in 2022?

Vegetation cover (VC) was measured monthly from 2019 to 2022 during the growing seasons using a steel wire frame (50 cm × 50 cm) subdivided into 25 plots of 10 cm × 10 cm. The cover of bare land (in the resubmitted manuscript, "bare land" will be replaced by "severely degraded meadow") has not changed significantly, and the bare land plots in Figure 1 of the manuscript were captured in 2022. We will provide a diagram with the values of vegetation cover per year in the the resubmitted manuscript.

Figure 2 and 3: In general, the illustrations are a bit overloaded with information. Therefore, I suggest to remove the jitter points and instead indicate the number of measurements per year, which should be the same for all treatments. Furthermore, I would remove the written mean values because they are already mentioned in the text. It would also be a good idea to choose different colours for your treatments, as they are indistinguishable to readers with colour vision deficiencies. Please also explain your boxplots in more detail in the figure caption, e.g., the line inside boxplots sometimes refers to median, sometimes to mean values, which is not explained here.

We will delete the jitter points, illustrate the number of measurements per year, erase the written mean values, and choose different colours for the different treatments in Figures 2 and 3.

[Figure]

**Figure 2.** Changes in soil erosion and runoff under various mixed-cultivated grasslands from 2019 to 2022. (a) Runoff depth and (b) soil erosion module. Note: For the four treatment runoff plots, runoff and sediment were measured 14, 18, and 10 times, respectively, during the growing season of 2019, 2020, and 2022. Different capital letters mean that differences were significant in different years for the same grassland community, and different lowercase letters mean that differences were significant between different communities in the same year. *SDM*, severely degraded meadows, *DE*, *Deschampsia cespitosa* and *Elymus nutans*; *PE*, *Poa pratensis L.cv.* Qinghai and *Elymus nutans*; and *PD*, *Poa pratensis L.cv.* Qinghai and *Deschampsia cespitosa*. The lines in the middle of the box represent the median values. The squares in the box represent the average value.

[Figure]

Figure 3. Runoff, soil loss and sediment concentration reduction ratio under different mixed-cultivated grasslands from 2019 to 2022. (a) Runoff reduction ratio (*RRE*), (b) soil loss reduction ratio (*SRE*), (c) sediment concentration reduction ratio (*CRE*) and (d) the percent of runoff reduction ratio to soil loss reduction ratio (*RRSR*). Note: Different capital letters mean that differences were significant in different years for the same grassland community, and different lowercase letters mean that differences were significant between different communities in the same year.

Figure 3: Please change the y-axis label of part (d) to "RRSR".

We will replace the y-axis label of part (d) "RRSE" by "RRSR" in the resubmitted manuscript.

Lines 232-234: Please explain how you came to this conclusion based on the results.

We will rewrite the sentence as follows: "The results showed that three mixed-cultivated grasslands (*DE*, *PE*, and *PD*) could be effective in controlling soil loss and maintaining runoff."

Lines 244-245: This sentence rather belongs to the discussion section.

We will erase this sentence in the resubmitted manuscript.

Line 247: Space is missing between "were" and "PD".

This will be corrected in the resubmitted manuscript.

Lines 251-252: This was only the case in 2022. From 2019 to 2020 the RRSR was higher than 1, is this right?

I am sorry maybe a misunderstanding exists. As shown in Fig. 3d, the mean *RRSR* values of the cultivated-grassland communities *DE*, *PE*, and *PD* were 30.3%, 29.5% and 22.8% in 2019, 20.0% 61.6%, and 62.0% in 2020, -26.0% -105.7%, and -132.2%, respectively.

Lines 254-264: Instead of listing all the values of the path analysis, it would be more comprehensible to list only the values of the most important parameters.

Thank you for your valuable suggestion. To be more understandable to the reader, we will only list the values of the most important parameters in the resubmitted manuscript.

Table 1: Please include the explanation of "*" in Table 1 instead of Table 2.

We will add the explanation of "*" in Table 1 and erase its explanation in Table 2 in the resubmitted manuscript.

Table 1 and 2: Sometimes the description of parameters in the caption is not clear, e.g., "ARI is average intensity" should be average rainfall intensity. Please check and clarify.

We have checked the whole paper and the words "average intensity" will be replaced by "average rainfall intensity" in the resubmitted manuscript.

Line 265: Please change to "with R being the most relevant".

This will be updated in the resubmitted manuscript.

Line 271: In Table 2 the indirect path coefficient of LB is -0.02, in the text it is -0.03. Please check again.

We have carefully checked the data processing and will correct the data in Table 2 and list the values of the relatively important parameters in the resubmitted manuscript.

**Discussion**

Lines 275-276: What do you mean by "conserve water"? Perhaps retaining or storing water? Also, it is not clear to me how you conclude that soil loss was minimized, since soil erosion on bare land was lower than on grassland in 2019 and 2020 after all, and in 2022 the difference was not significant (Figure 2b).

The word "conserve water" will be replaced by "maintain runoff". The sediment concentration reduction ratio (CRE) and soil erosion reduction ratio (SRE), which reflected reductions in soil loss, continue to increase from 2019 to 2022. These results supported our conclusions.

Lines 291-293: Are these percentages based on the average values from Table 3? If this is the case, I assume, the percentages are not correct.

Maybe there has been a misunderstanding. We have carefully checked the data processing and did not find mistakes. Yes, the increase rate of root mass density and soil cohesion was obtained based on the average values from Table 3. Increase rate = $(V_{MCG}-V_{SDM})/V_{SDM}$), where $V_{SDM}$ and $V_{ACG}$ are the mean values of root mass density and soil cohesion of three mixed-cultivated grasslands (*DE*, *PE*, and *PD*) and severely degraded meadow (*SDM*), respectively.

Lines 301-302: I do not really understand what is meant by this sentence.

We will revise the sentence in the resubmitted manuscript as follows: "Surface runoff and erosion process is influenced and constrained by rainfall depth, intensity and duration, and by vegetation cover as well (Mohamadi and Kavian, 2015b; Bochet et al., 2006)."

Line 321: Please correct the spelling of Poa pratensis here and in the whole paragraph.

This should have been "*Poapratensis*" to "*Poa pratensis*" and will be corrected in the resubmitted manuscript.

Lines 322-323: It is unclear to me how you come to this conclusion, since there were no significant differences between the grass mixtures.

Thank you for this comment. We will revise the too strong statement as follows: "When the mixed-cultivated grasslands could act as a consolidation effect on the surface soil (the 4th year of planting after tilling), the community of *PD* (0.12 g m$^{-2}$) was probably more effective than the communities of *PE* (0.18 g m$^{-2}$) and *DE* (0.16 g m$^{-2}$) in reducing soil loss (Fig. 3), which could likely be due to two

reasons."

Lines 348-352: This part rather belongs to the introduction section to underline the importance of this study.

We will erase this part in the discussion section in the resubmitted manuscript.

Line 353-354: I suggest deleting the term "Overland flow turbidity" here, as this is the first time it has been used and it is confusing when new terms are introduced at the end of the discussion.

The term "Overland flow turbidity" will be deleted in the resubmitted manuscript.

**Conclusions**

Lines 356-359 + 361-364: These are too strong statements for the data you presented in your manuscript.

We will revise the too strong statement as follows: "Based on the measured data during the 2019, 2020 and 2022 growing seasons, the planting of mixed-cultivated grassland on the severely degraded hillside alpine meadow could effectively maintain surface water and decrease soil loss, especially because the mixed-cultivated grassland played a positive role in consolidating the surface soil."

**References:**

Bochet, E., and García-Fayos, P.: Factors controlling vegetation establishment and water erosion on motorway slopes in Valencia, Spain, Restor. Ecol., 12(2), 166–174, https://doi.org/10.1111/j.1061-2971.2004.0325.x, 2004.

Bochet, E., Poesen, J., and Rubio, J.L.: Runoff and soil loss under individual plants of a semi-arid Mediterranean shrubland: influence of plant morphology and rainfall intensity. Earth Surf. Proc. Land 31, 536–549, https://doi.org/10.1002/esp.1351, 2006.

De Baets, S., Poesen, J., Knapen, A., Barberá, G.G., and Navarro, J.A.: Root characteristics of representative Mediterranean plant species and their erosion-reducing potential during concentrated runoff, Plant Soil, 294(1–2), 169–183, https://doi.org/10.1007/s11104-007-9244-2, 2007.

Huang, Z., Liu, Y.F., Cui, Z., Liu, Y., Wang, D., Tian, F.P., and Wu, G.L.: Natural grasslands maintain soil water sustainability better than planted grasslands in arid areas. Agr. Ecosyst. Environ., 286(1), 106683, https://doi.org/10.1016/j.agee.2019.106683, 2019.

Huang, Z., Tian, F.P., Wu, G.L., Liu, Y., and Dang, Z.Q.: Legume grasslands promote precipitation infiltration better than gramineous grasslands in arid regions, Land Degrad. Dev., 28(1), 309–316, https://doi.org/10.1002/ldr.2635, 2017.

Li, W., Wang, J.L., Zhang, X.J., Shangli, S., and Wenxia, C.: Effect of degradation and rebuilding of artificial grasslands on soil respiration and carbon and nitrogen pools on an alpine meadow of the Qinghai-Tibetan Plateau, Ecol. Eng., 111, 134–142, https://doi.org/10.1016/j.ecoleng.2017.10.013, 2018.

Liu, Y., Zhao, L.R., Liu, Y.F., Huang, Z., Shi, J.J., Wang, Y.L., Ma, Y.S., Lucas-Borja, M.E., L´opez-Vicente, M., and Wu, G.L.: Restoration of a hillslope grassland with an ecological grass species (*Elymus tangutorum*) favors rainfall interception and water infiltration and reduces soil loss on the Qinghai-Tibetan Plateau, Catena, 219, 106632, https://doi.org/10.1016/j.catena.2022.106632, 2022.

Liu, Y.F., Liu, Y., Wu, G.L., and Shi, Z.H.: Runoff maintenance and sediment reduction of different grasslands based on simulated rainfall experiments., J. Hydrol., 572, 329–335, https://doi.org/10.1016/j.jhydrol.2019.03.008, 2019.

Ren, F., Zhou, H.K., Zhao, X.Q., Han, F., Shi, L.N., Duan, J.C., and Zhao, J.Z.: Influence of simulated warming using OTC on physiological–biochemical characteristics of Elymus nutans in alpine meadow on Qinghai-Tibetan plateau, Acta Ecol. Sinica, 30(3), 166–171, https://doi.org/10.1016/j.chnaes.2010.04.007, 2010.

Shi, X.N., Zhang, F., Wang, L., Jagirani, M.D., Zeng, C., Xiao, X., Wang G.X.: Experimental study on the effects of multiple factors on spring meltwater erosion on an alpine meadow slope. International Soil and Water Conservation Research, 8(2): 116-123. https://doi.org/10.1016/j.iswcr.2020.02.001, 2020.

Zhang, G.H., Tang, K.M., Ren, Z.P., and Zhang, X.C.: Impact of grass root mass density on soil detachment capacity by concentrated flow on steep slopes, T. ASABE, (56), 927–934, 2013.

Zheng, Y., Shi, X. N., Zhang, F., Lei, T. W., Zeng, C., Xiao, X., Wang, L., and Wang, G. X.: Field experimental study on the effect of thawed depth of frozen alpine meadow soil on rill erosion

by snowmelt waterflow. *International Soil and Water Conservation Research*, in press. https://doi.org/10.1016/j.iswcr.2022.12.001, 2022.

---

## Author Comment (AC3)

**Initial response to RC2**

*Veerle Vanacker (Reviewer 2)'s original text in black with our initial response in* blue.

This manuscript describes the results of empirical research on the effectiveness of three different restoration measures in a grassland ecosystem. Given that native grasslands are increasingly subject to degradation, such empirical work can be very relevant for restoration efforts. The strength of this work lies in the collection of empirical data on surface runoff and soil loss rates from 4 runoff plots that were monitored over three years. As there are no replicates, the evaluation of the effectiveness of the treatments is based on a time series of events.

Given a longstanding interest in soil and water conservation measures, and their effectiveness and efficiency, there exists a vast amount of literature on the topic. It includes not only empirical work on e.g. runoff or Wishmeier plots, but also regional synthesis on the effectiveness of conservation measures. This manuscript would be strengthened by embedding it better in the international literature, using e.g. the terminology that was established in soil erosion research, and using standardised methods for measuring the effectiveness of treatments.

Four major points caught my attention, and they might guide the revisions. Besides these points, I have some detailed comments that are listed below.

We thank Professor Veerle for her positive comments and for the critical and constructive suggestions. We believe that the reviewers addressed important points and that these comments allowed us to further clarify and strengthen the manuscript.

1-The methodological aspects need to be better explained to the reader as to avoid confusion. The effect of the measures is quantified through "reduction ratios" of runoff, soil erosion and sediment concentration. Is it not entirely clear why the authors have chosen to focus on the "reduction ratios" rather than the absolute values of runoff quantity, and soil erosion rate? What is the added value of introducing such ratios? What is the difference between the effect on soil erosion, and the effect on the "sediment concentration" if both are quantified based on measurements from the sediment that is eroded from the plots, and captured in the tank? Given that the size of the plots is the same, the

"sediment concentration" measure would be redundant if the "runoff depth" and "soil erosion rate" were given. Can you explain why you keep "concentration" as an indicator?

Runoff reduction ratio, sediment concentration reduction ratio, soil erosion reduction, could directly reflect the efficiencies of the three mixed-cultivated grasslands in maintaining runoff and reducing soil erosion (Zhao et al., 2014; Zhu et al., 2021). The study area is located in a headwater catchment, supplying significant freshwater resources to the Mekong, Yangtze and Yellow rivers. Hence, the primary objective of restoration measures is to effectively maintain runoff while minimizing its concentration. The measurement of sediment concentration serves as a crucial indicator for assessing water conservation practices in the study area.

I can see that the authors want to compare treatments, with the "control" of bare land, but are the experimental plots in a comparable initial state? From the text, one might have the impression that the "control" site is left untouched while the other plots have been prepared for seeding, including ploughing of the site. Ploughing might have a strong impact on the soil physical properties, and change surface roughness, break up soil crusts and enhance infiltration. As such, there is an effect of ploughing and then of the vegetation. If the "control" plot is not ploughed, how do you differentiate between the effect of mechanical ploughing (and breaking up soil crusts) and the effect of vegetation?

Thank you for this valuable comment for further research. Ploughing has a significant effect on the soil physical properties, and changes surface roughness, breaks up soil crusts and enhances infiltration. We employed bare land (in the resubmitted manuscript, "bare land" will be replaced by "severely degraded meadow"), as control to quantify the increase in soil erosion within the mixed-cultivated grassland compared to the control during the initial planting stages. Additionally, we evaluated the effectiveness of the mixed-cultivated grassland in maintaining runoff and reducing soil erosion when it exerted soil consolidation in the unrecovered degraded grasses. We will add this explanation in the revised version of the manuscript.

2-There is some confusion in the text with regard to vegetation restoration as a measure to control water runoff and soil erosion. This can be due to the framing of the research, and the use of terminology. For example, the authors often refer to "control runoff and sediment" but it would be good to have more precise use of the terminology that is commonly used in soil erosion studies. Are

the authors referring to "in-situ runoff" and "soil erosion"? or "sediment mobilisation and lateral transport"? In the methods (L184 and following), the authors also introduce another term to express the soil loss rate, which is the "soil erosion modulus". It is not clear to me why a new term is introduced.

To avoid confusing the reader, we will modify the use of terminology in the resubmitted manuscript, such as "soil erosion" instead of "sediment", as well as "soil erosion amount" instead of "soil erosion modulus".

3-The authors describe the soil and water conservation treatments in different ways, which is somehow confusing for the reader. A clear description of the vegetation composition of the alpine grasslands would be helpful, as well as an explanation of the human alterations of these grasslands. What are "artificial grasslands", "mixed artificial grasslands", mixed cultivated grasslands"? What are the differences between them in plant species composition?, How are "artificial grassland restoration" projects (L106) done? In the methods, the experiments are rather well described, but it remains unclear if the grasses that are used in the restoration project are native grasses if they are common/abundant in the region, and if the plant composition (two grass species) is common in the region.

Also, it is necessary to describe the initial state of the plots (see comment above). What are the soil physical properties of the plots before running the experiments? Do you have information on bulk density, soil crusts, and topographic roughness? Are these properties similar between the plots?

Thank you for your valuable suggestions. The words "artificial grassland", "mixed artificial grassland" and "mixed cultivated grassland" in the manuscript refer to mixed-cultivated grassland. To avoid confusion for the readers, we will revise them uniformly to "mixed-cultivated grassland" in the resubmitted manuscript. The projects of artificial grassland restoration have been carried out in 2019 via mixing grass seeds after plowing on the severely degraded alpine hillside. We will provide a detailed illustration in the resubmitted version of the manuscript.

All four runoff plots are situated on the same hillside at the same elevation. As a result, the topsoil properties in the bare land (in the resubmitted manuscript, "bare land" will revise to "severely

degraded meadow") plots in Table 3 can represent the topsoil properties of the three mixed cultivation meadow plots prior to plowing.

4-In the description of the results, there is a tendency to overstate the results (and the effect of the treatments). The authors have realised an ANOVA analysis (with posthoc comparison->?) to verify if the treatments lead to different runoff or soil erosion rates (or different ratios). As there are no replicate plots/samples, the authors take the response to individual events as their observations. Wouldn't you expect that the response to these events is somehow autocorrelated? Is there a way to deal with this? It also makes the analyses very sensitive to the definition of the events, which makes me wonder how events were defined/delimited in time. This would merit further explanation in the methods, including a discussion of how consecutive events are used as independent observations.

The description of the results was indeed slightly overstated because the differences were not significant, but the three mixed-cultivated grasslands did exhibit a clear reduction in soil loss in 2022 compared to the bare ground (in the resubmitted manuscript, "bare land" will appear as "severely degraded meadow"). We will revise the description of the results from Figs. 2 and 3 in the resubmitted manuscript. In this experiment, precipitation events were determined by the presence of a no-rain period between them longer than 3 h. This will be updated in the resubmitted manuscript.

From Figures 2 and 3, it is clear that the three interventions lead to responses that are often statistically different from the "control" bare land, but that there is rarely a statistical difference between the interventions (they are most of the time all part of the "a" group in Fig 2 and 3). The authors might need to revise the text (section 3.1, and section 3.2) to highlight only significant differences. Also, with regard to the ratios, the text needs to be revised after verifying that the responses are statistically different from 1.

Thank you for noticing this. As a result, we will carefully check the letters in Figures 2 and 3 and modify the expressions of the results in those figures in the resubmitted manuscript. In our opinion, the presence of clear differences, although not significant, also represents a relevant result that gives value to our study.

The path analyses merit to be further explained in the method section. What are the variables that go into the path analyses, and what are the observations? In the results, the authors mention that

"vegetation cover" and "litter biomass" are explanatory factors, but it is not clear to me how the authors compared the information that you collect at event basis with information that is collected at the start/end of the experiments (VC and LB). Also, some variables like the precipitation (P) would be automatically related to the runoff depth for a given treatment. This merits further explanation and development in the text to understand the results of the path analysis presented in section 3.3.

The runoff depth and soil erosion modulus were response variable, and explanatory variable included maximum 60-minute intensity, average rainfall intensity, rainfall duration, rainfall amount, vegetation coverage and litter biomass. Despite some factors having a correlation, the path analysis was unaffected by this. For instance, there is a correlation between rainfall and runoff depth, thus rainfall directly impacts the soil loss (direct path coefficient), but it also indirectly affects the amount of soil loss through runoff depth (indirect path coefficient). To provide the reader with a better understanding of the path analysis results, the path analyses merit and the methods of vegetation cover and plant litter biomass will be added in the resubmitted manuscript.

Detailed comments

L10: check "4" instead of "d"
This should have been "4" and will be updated in the resubmitted manuscript.

L18: Please rephrase "control runoff and sediment (?)". Do you mean sediment transport?

The word "sediment" in the manuscript refers to "soil erosion". We will modify this sentence in the resubmitted manuscript.

L19: "objectively" is not the correct term. do you suggest that previous work was subjective?

We will erase the word "objectively" in the resubmitted manuscript.

L22: The effectiveness of measures is often scale-dependent. What is the size of the plots?

The runoff plots have a size of 10 m$^2$, with a width of 2 m and a length of 5 m. We will erase the word "effectiveness" in the resubmitted manuscript.

L25: What do you mean by "conserve water"?

To avoid confusing the reader, the words "conserve water" will revise to "maintain runoff" in the resubmitted manuscript.

L46: Can you rephrase this sentence? Do you mean that the highest population concentrations on earth are found in grassland ecosystems?

We will modify this sentence in the resubmitted manuscript.

L51: Can you check the wording "analyse...root causes, .. impacts and restoration measures of grassland degradation" doesn't read very well.

We will modify this sentence in the resubmitted manuscript.

L58: Efficiency or effectiveness? Please check.

This should have been "effectiveness" and will be updated in the resubmitted manuscript.

L64: Can you clarify this sentence? What do you mean by "enhancement of soil characteristics with the growth of vegetation"? Which soil properties are improved after vegetation restoration?

Vegetation restoration could encourage the formation of soil aggregates, particularly water-stable aggregates, and increases the soil organic matter content via root growth (Liu et al., 2020; Saxton and Rawls, 2006). We will add a description and references on improved soil properties after vegetation restoration.

L73: Please check the wording of this and the following sentence. The terminology is not always used in the correct way. What do you mean by "grasses below-ground"? Or with the "interaction of soil and rich grassroots"?

This is a typo, and the sentence will appear as "Soil erosion can decrease with grasses above- and below-ground biomass and litter cover, as well as root systems" in the resubmitted manuscript. In addition, the reciprocal cementation and interweaving of plant roots improved soil structure by promoting the formation and stability of large aggregates. The words "interaction of soil and rich grassroots" will be replaced by the words "the reciprocal cementation and interweaving of plant roots".

L83: What is the effect of the roots on the soil mass? Can you be more explicit here?

Plant roots have the effect of loosening the soil, resulting in the increase of soil pores and the decrease of soil bulk density (Gyssels et al., 2005; Wu et al., 2019). We will provide explanation in the resubmitted manuscript.

L87: Can you check the wording? "alpine grasslands" are not a plant type.

Thank you for noticing this. This should have been "alpine meadow" and will be updated in the resubmitted manuscript.

L90: What do you mean by "non-planned human activities"?

We will specify "non-planned human activities" as "overgrazing" in the resubmitted manuscript.

L87-98: Given the overall concern of increased soil loss in the alpine grasslands, can you give some quantitative data on soil loss rates in the area (and compare them with other regions)?

The study area is characterized for the shallowness of the soil layers, stony soils beneath mattic layer and low-intensity rainfalls, which led to a lower runoff depth and soil erosion modulus of alpine grasslands. Soil erosion in the study area was mainly mild (83.83% of the total eroded area), and the average soil erosion rate and the total erosion were 13.63 t ha$^{-1}$ y$^{-1}$ and 323.58 × 10$^6$ t y$^{-1}$ respectively, before implementation of the program 'Subsidy and Incentive System for Grassland Conservation' (Zhao et al., 2021). However, the cold and harsh environment of the Plateau and the slow rate of microbial decomposition have led to weak soil development, slow soil formation and very thin soil layers in the study area, and thus soil erosion must be really controlled. We will add this information to the introduction section in the resubmitted manuscript.

L114: Can you be more specific? What do you mean by "just cold and warm'?

In the study region, the average annual temperature is -0.1°C, with monthly variations from -18.3 °C in January to 12.4 °C in July (Li et al., 2018). We will add specific information in the resubmitted manuscript.

L122: Can you give the species that were used in restoration efforts?

Thank you for your valuable suggestions. The grass species used for grassland restoration projects have excellent characteristics like strong resistance to stress, rich leaf quantity, good palatability, developed rhizome and strong grazing resistance, such as *Poa pratensis L. cv.* Qinghai and *Elymus nutans* (Shang et al., 2018). They are of rhizomatic type, so they can quickly form turf on degraded alpine hillslope, restore vegetation and stabilize soil surface. We will provide the species names that were used in the restoration efforts in the resubmitted manuscript.

L127: Can you be more specific about the meaning of "concentration areas"?

This should have been "confluence areas" and will be updated in the resubmitted manuscript.

L129: alpine climate (?)

The word "climatic" will be replaced with "climate" in the resubmitted manuscript.

L131: Can you reword "blending complementary grass species"?

The words will reword to "matching of grasses morphological characteristics and habits".

L137: above-ground "biomass"? instead of "plants"

We think the words "grass stems" are more appropriate.

L150-154: Are there any replicate plots?

The runoff plots were not replicated.

L170: In a conventional rain gauge, snow would also fall in the pluviometer. So, did you account/correct for that?

Yes, snow falls were accounted into the rain gauge. It is worth saying that in this study, we only monitored rainfall events during the growing season (rainy season), and no snowfall occurred during that period.

L172: Average intensity of the rain event? or of the 60-minute interval?

Average intensity refers to the average rainfall intensity of each rainfall event.

L178: Can you check the wording: what is "qualitative filter paper"?

Thank you for noticing this. This should have been "quantitative filter paper" and will be updated in the resubmitted manuscript. When referring to "qualitative analysis filter paper", qualitative analysis filter paper is relative to quantitative analysis filter paper and chromatographic qualitative analysis filter paper. Qualitative filter paper is a kind of paper with good filtering performance, loose texture, and strong liquid absorption capacity. Quantitative filter paper is mainly used for ashing and gravimetric analysis experiments after filtration. That is gravimetric analysis tests and corresponding analysis tests in quantitative chemical analysis.

L178-179: There is some inconsistency here: either air-dried or dried at 105°C. Please check.

This should have been "oven-dried" and will be updated in the resubmitted manuscript.

L197: Please check the wording.

Soil shear strength was tested using a direct shear (ZJ type). Then, soil cohesion was calculated from the equation of the Mohr-Coulomb line of the soil model. The sentence will be revised to "The soil cohesion was calculated from the equation of the Mohr-Coulomb line based on direct shear test".

L217: Not entirely clear how this is conceived with 4 plots.

One plot with severely degraded meadow as a control and three plots with mixed-cultivated grasslands, namely *Deschampsia cespitosa* and *Elymus nutans* (*DE*), *Poa pratensis L.cv.* Qinghai and *Elymus nutans* (*PE*), and *Poa pratensis L.cv.* Qinghai and *Deschampsia cespitosa* (*PD*) (Fig. 1). We employed severely degraded meadow as a control to quantify the increase in soil erosion within the mixed-cultivated grassland compared to the control during the initial planting stages. Additionally, we evaluated the effectiveness of mixed-cultivated grassland in maintaining runoff and reducing soil erosion when it exerted soil consolidation compared to the unrecovered degraded meadow. We will add this explanation to the revised version of the manuscript.

L223: Based on the observations, you cannot conclude that the vegetation restoration increased runoff. In fact, you compare 4 plots with different vegetation cover and different levels of soil roughness and sealing. To conclude that the vegetation restoration has increased the runoff, you would need to do a before/during/after evaluation of the effectiveness of the treatment.

Maybe a misunderstanding existed. All four runoff plots are located on the same hillside at the same elevation and soil texture. The interval between plots is 1 m. Observations of severely degraded grassland were comparable to those made prior to treatment; those obtained in 2019 and 2020 of mixed-cultivated grassland were comparable to those made during the treatment; and those obtained in 2022 of the mixed-cultivated grassland were comparable to those made following the treatment. To avoid confusing the reader, we will add an illustration in section 2.2.

L222 and following: This conclusion "grass communities dramatically influence runoff" is not fully supported. There is a difference between the runoff depth between the bare land and the restoration plots, in 4 out of 6 cases. No difference between the other treatments.

"Grass communities dramatically influenced" should have been "Mixed-cultivated grasslands dramatically increased" and will be updated in the resubmitted manuscript.

L238: Can you revise the wording of this sentence? What do you mean by "surface water conservation"?

To avoid confusing the reader, the words "surface water conservation" should have been revised to "maintaining runoff".

L237-252: Based on the equations provided, the effectiveness is measured using ratios but in the text, the author refers to %. Please check for consistency.

We will modify the formulas (4)-(6) in the resubmitted manuscript.

L280 and the following: What do you mean by "soil erosion modulus"?

We will replace "soil erosion modulus" with "soil erosion amount".

Section 4.3: The last section of the discussion deals with the "implications of grassland restoration", and discusses differences in surface runoff and soil loss between different treatments. Are the differences significant?

The differences in surface runoff and soil loss between the three mixed-cultivated grasslands were not significant. Hence, we will modify the expression related to this in the resubmitted manuscript.

L356-359: Is this what you observe in your results when you compare the "treated" plots with the "control" plots?

Based on the measured data during the 2019, 2020 and 2022 growing seasons, the planting of mixed-cultivated grassland on severely degraded hillside alpine meadow could effectively maintain surface water and decrease soil loss, especially because the mixed-cultivated grassland played a positive role in consolidating the surface soil. We will modify the text in the resubmitted manuscript.

Figure1: Nice figure especially the pictures on the runoff plots. Is it possible to show the scalebar on all maps, and give some reference on the spatial scale of the pictures?

We will add the size of the runoff plots on Figure 1 in the resubmitted manuscript.

Figure 3: The reduction ratios are mentioned as fractions in the method sections, which should be the case for a "fraction", but they are then displayed as % in Figure 3. Please check.

Thank you for noticing this. We will modify the formulas (4)-(6) in the method sections in the resubmitted manuscript.

Tables 1 &2: Some background info on the path analysis would be welcome. You could add a few sentences in the methods to explain the background of this stat analysis.

Path analysis is a form of multiple regression statistical analysis that is used to evaluate causal models by examining the relationships between a dependent variable and two or more independent variables. By using this method, one can estimate both the magnitude and significance of causal connections between variables. In this study, the path analysis method was utilized to determine the main factors influencing runoff and soil erosion. We will add the background information and merit of path analysis in the resubmitted manuscript.

References

Gyssels, G., Poesen, J., Bochet, E., and Li, Y., Impact of plant roots on the resistance of soils to erosion by water: a review, Progr. Phys. Geogr., 29(2), 189–217, https://doi.org/10.1191/0309133305pp443ra, 2005.

Li, W., Wang, J.L., Zhang, X.J., Shangli, S., and Wenxia, C.: Effect of degradation and rebuilding of artificial grasslands on soil respiration and carbon and nitrogen pools on an alpine meadow of the Qinghai-Tibetan Plateau, Ecol. Eng., 111, 134–142, https://doi.org/10.1016/j.ecoleng.2017.10.013, 2018.

Liu, Y., Guo, L., Huang, Z., López-Vicente, M., Wu, G.L.: Root morphological characteristics and soil water infiltration capacity in semi-arid artificial grassland soils. Agr. Water Manage. 235, 106153. https://doi.org/10.1016/j.agwat.2020.106153. 2020.

Saxton, K.E., Rawls, W.J.: Soil water characteristic estimates by texture and organic matter for hydrologic solutions. Soil Sci. Soc. Am. J., 70(5), 1569–1578. https://doi.org/10.2136/sssaj2005.0117. 2006.

Shang, Z., Dong, Q., Shi, J., Zhou, H., Dong, S., Shao, X., Li, S., Wang, Y., Ma, Y., Ding, L., Cao, G., Long, R.: Research progress in recent ten years of ecological restoration for 'black soil land' degraded grassland on Tibetan plateau-concurrently discuss of ecological restoration in Sanjiangyuan region. Acta Agrestia Sinica 26, 1–21. https://doi.org/10.11733/j.issn.1007-0435.2018.01.001. 2018.

Wu, G.L., Huang, Z., Liu, Y.F., Cui, Z., Shi, Z.H.: Soil water response of plant functional groups along an artificial legume grassland succession under semi-arid conditions. Agr. Forest Meteorol., 278, 107670 https://doi.org/10.1016/j.agrformet.2019.107670. 2019.

Zhao, X., Huang, J., Wu, P., Gao, X.: The dynamic effects of pastures and crop on runoff and sediments reduction at loess slopes under simulated rainfall conditions. Catena, 119, 1–7. https://doi.org/10.1016/j.catena.2014.03.001. 2014.

Zhao, Y.T., Pu, Y.F., Lin, H.L., Tang, R.: Examining soil erosion responses to grassland conversation policy in Three-River Headwaters, China. Sustainability, 13, 2702, https://doi.org/10.3390/su13052702. 2021.